# Towards Human-Like Event Boundary Detection in Unstructured Videos through Scene-Action Transition

## Abstract

Humans segment continuous experience into episodes by detecting perceptual novelties and retrospectively consolidating them into coherent memories. Drawing inspiration from these cognitive principles, we introduce a two-level, backward-only event segmentation framework designed to structure continuous sensory input into stable episodic units. The goal is to identify meaningful situational shifts (e.g., transitions between activities or environments) that lie between coarse scene-change detection and the dense, motion-driven micro-boundaries targeted in Generic Event Boundary Detection (GEBD). At Level 1, an error-driven novelty detector with semi-supervised adaptive thresholding identifies candidate transitions robust to noise, viewpoint shifts, and repeated micro-actions. At Level 2, a retrospective boundary consolidation mechanism validates and merges these candidates using multimodal cues (scene graphs, captions, audio), producing stable, semantically grounded episodes without relying on future frames. Unlike prior GEBD approaches that depend on motion cut-points or heavy task-specific supervision, our method uses sparse labels only for threshold calibration, making it label-efficient, cognitively inspired, and broadly applicable. Experiments on Ego4D show state-of-the-art performance, with our semi-supervised model surpassing heavily supervised baselines. This work introduces episodic segmentation for embodied perception, taking conceptual inspiration from human memory research while focusing on scalable machine perception.

## 1 Introduction

Humans naturally parse continuous experience into events, a process known as event segmentation (Nguyen et al., 2025; Michelmann et al., 2023b;a). Boundaries are perceived when perceptual features (e.g., motion, sound) or conceptual features (e.g., goals, intentions) change, forming a hierarchical structure of fine- and coarse-grained episodes. These boundaries are not arbitrary: they scaffold episodic memory, enabling people to recall past experiences, learn new skills, and anticipate future outcomes.

Figure 1 illustrates our dual-level framework: error-driven novelty detection is retrospectively consolidated into semantically coherent episodes.

Inspired by these ideas, we ask: *How can an artificial agent segment its continuous sensory stream into meaningful episodes suitable for episodic memory?* Unlike offline video analysis, an embodied agent must structure experience in real time based on places, participants, and task-level transitions rather than superficial discontinuities.For example, in Activities of Daily Living (ADL), relevant transitions include entering a new room, shifting from preparing to cooking, or the arrival of a new person—precisely the type of semantic boundaries that support long-horizon memory and reasoning.

A cognitively capable agent should therefore segment memory into stable, interpretable, and semantically meaningful episodes. Such segmentation enables agents to compress continuous experience, support causal reasoning, and maintain long-term coherence. In contrast, existing GEBD methods often rely on motion-driven cut points, which fragment continuous streams and degrade memory stability.

We propose a cognitively grounded representation learning framework for event segmentation. Our central idea is that boundaries should emerge not from immediate motion changes, but from the ret-

Figure 1: Our cognitively inspired hierarchical event segmentation framework. Level 1 detects fine-grained perceptual novelties while Level 2 retrospectively consolidates boundaries into stable, semantically coherent episodes (*retrospective boundary consolidation*). Motion-driven GEBD methods often fragment actions into multiple cuts—e.g., each knife jitter during sandwich preparation may be marked as a separate boundary. In contrast, our approach groups such micro-changes into a single meaningful action (e.g., *spreading butter on bread*), providing more coherent episodes.

rospective stability of representations over time. To achieve this, we introduce a *backward-looking temporal windowing mechanism* that compares the present to the recent past, avoiding reliance on unavailable future frames. At a second level, we retrospectively consolidate candidate boundaries using scene graphs, audio cues, and caption semantics, ensuring that episodes reflect stable shifts in meaning rather than transient visual changes. In addition, we introduce a *semi-supervised adaptive thresholding module* that learns to calibrate novelty sensitivity from retrospective statistics, improving robustness to noise, jitter, and viewpoint shifts.

**Key Contributions**

- **Cognitively grounded paradigm for cognitive agents:** We propose a dual-level, backward-only event segmentation framework inspired by human episodic memory, enabling artificial agents to structure continuous sensory streams into interpretable episodes.

- **Semi-supervised adaptive threshold detection:** A label-efficient thresholding mechanism that dynamically calibrates sensitivity from retrospective statistics, improving robustness to noise, jitter, and viewpoint changes.

- **Multimodal integration:** Our approach consolidates boundaries using semantic (captions, scene graphs), perceptual (DINOv2, SSIM, LPIPS), and linguistic (dialogue-aware) cues in a unified retrospective validator, without reliance on dense frame-level labels or task-specific fine-tuning.

- **Strong empirical validation:** Experiments on ADL-GEBD and Ego4D indicate that our semi-supervised framework achieves performance comparable to motion-driven GEBD methods and supervised models, suggesting good scalability and semantically coherent segmentation.

## 2 RELATED WORK

**Generic Event Boundary Detection (GEBD).** Generic Event Boundary Detection (GEBD) Mike Zheng Shou & Feiszli. (2021) aims to localize perceptual transitions in video without predefined labels. Early methods formulated GEBD as frame-level binary classification Mike Zheng Shou & Feiszli. (2021); Jiaqi Tang & Wang. (2022); Dexiang Hong & Zhang. (2021), but these models often over-segment due to their reliance on superficial appearance or motion changes. More recent work incorporated contrastive learning Hyolim Kang & Kim. (2021), compact encodings Congcong Li & Zhang. (2022), and transformer-based architectures Sourabh Vasant Gothe & Kashyap. (2023); Congcong Li & Wen. (2022), often coupled with optical flow Rui Qian & Cui. (2021). While effective at detecting local visual novelty, such approaches tend to produce fragmented segmentations that struggle with higher-level semantics such as goals or dialogue continuity. Unsupervised variants (e.g., PySceneDetect Castellano., PredictAbility Mike Zheng Shou & Feiszli. (2021), CoSeg Xiao Wang & Luo. (2109)) exploit

Table 1: Comparison of GEBD objectives versus our episodic segmentation task. GEBD emphasizes sensitivity to fragmented shifts, while our task prioritizes stable, semantically grounded episodes.

| Aspect | GEBD Focus | Our Task Focus |
|---|---|---|
| Granularity | Micro changes and frame-level transitions (often motion- or appearance-driven) | Coarse, semantically coherent episodes (scene + action + dialogue continuity) |
| Output Style | Fragmented boundaries highlighting perceptual shifts and uncertainties | Stable, consolidated episodes preserving narrative and semantic flow |
| Strengths | Sensitive to subtle changes; effective at detecting ambiguous or uncertain regions | Captures long-horizon coherence; supports reasoning, memory, and downstream tasks |
| Limitations for Our Use Case | Over-fragmentation → splits continuous dialogue, micro-actions, or camera jitter into many segments | Possible under-segmentation if overly coarse, but maintains meaningful episodic units |
| Cognitive Alignment | Perceptual novelty and local frame changes | Episodic memory structure (what, when, where), retrospective consolidation |

reconstruction losses or pixel variations, while hybrids such as UBoCo Hyolim Kang & Kim. (2007) combine multiple objectives. Despite these advances, most GEBD approaches remain focused on micro-level granularity. As summarized in Table 1, this sensitivity makes GEBD well-suited for perceptual novelty detection, but misaligned with the stability required for episodic segmentation.

**Motion and Visual Correspondence Learning.** Motion cues have long been central to video understanding, from classical optical flow Lucas & Kanade. (1981); Farnebäck. (2003) to modern motion-aware architectures Heeseung Kwon & Cho. (2020); Jiaqi Tang & Wang. (2022); Ayush K Rai & O'Connor. (2023). These techniques are effective for dense action localization but often generate visually reactive segmentations that neglect semantic continuity. Our approach diverges by avoiding explicit motion cues, instead leveraging semantically aligned representations with adaptive thresholds that flexibly capture both fine and coarse boundaries—crucial in egocentric or dialogue-heavy videos where appearance shifts may not correspond to meaningful transitions.

**Egocentric Video and Multimodal Understanding.** The Ego4D benchmark Grauman et al. (2022) has driven progress in egocentric video research, emphasizing tasks such as episodic memory and natural language query (NLQ). Current solutions typically adopt proposal-based Mo et al. (2022) or transformer-based Lei et al. (2021) pipelines, built on pretrained vision–language encoders like CLIP Radford et al. (2021b;a), VideoMAE Tong et al., or InternVideo Chen et al. (2022b). While effective for fine-grained retrieval, these systems are optimized for short-term alignment and often fail to capture higher-order transitions, such as shifts across environments or narrative stages.

In contrast to GEBD (Table 1), which emphasizes sensitivity to micro changes, our work prioritizes stable, semantically coherent episodes. By integrating semantic representations with a learnable boundary threshold, our approach captures both fine and coarse transitions without over-fragmentation. This enables structured episodic understanding, which is particularly beneficial for applications in robotics, surveillance, and assistive systems where long-horizon coherence and memory alignment are essential.

## 3 APPROACH

Episodic memory encodes not only *what* happened, but also *when* and *where* it occurred (Tulving, 2002). For autonomous agents, this requires transforming continuous sensory streams—such as egocentric video—into stable, semantically coherent episodes. We propose a cognitively inspired two-level framework: (i) adaptive boundary detection, which selects candidate transitions based on retrospective statistics within a short backward window, and (ii) *retrospective consolidation*, which validates and merges boundaries into coherent episodes using semantic, perceptual, and dialogue cues. Both mechanisms are strictly *backward-facing*, reflecting the episodic memory constraint that only past context is available at decision time.

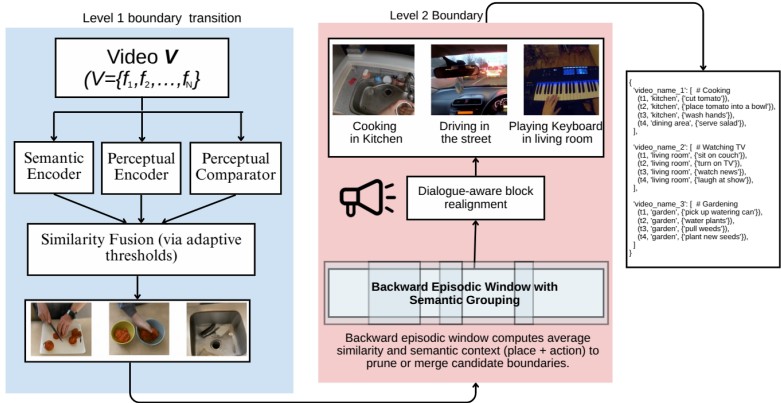

Figure 2: Overview of our cognitively inspired two-level episodic segmentation framework. **Level 1** (left, blue) identifies candidate boundaries through *adaptive thresholding*, comparing incoming frames against retrospective statistics within a fixed backward window across semantic, perceptual, and comparative encoders. **Level 2** (right, red) retrospectively validates and consolidates these candidates via *multimodal integration*, using semantic grouping (place + action), perceptual similarity, and dialogue alignment in a unified validator. Both stages are inherently *backward-facing*, operating only on past context to transform continuous sensory streams into stable, semantically coherent episodes of *what*, *when*, and *where*.

### 3.1 LEVEL 1: ADAPTIVE BOUNDARY DETECTION

The first stage proposes candidate boundaries by comparing each incoming frame with a fixed window of the $k$ most recent frames. Frames $F = \{I_1, I_2, \ldots, I_N\}$ are uniformly sampled, and a subset $K \subset F$ of keyframes is selected when local similarity drops below an adaptive threshold. Each frame $I_i$ is encoded via three parallel streams: a *Semantic Encoder* (objects, relations, and high-level concepts), a *Perceptual Encoder* (appearance and spatial structure), and a *Comparator* (patchwise dissimilarity). The fused similarity between frames $I_j$ and $I_i$ is defined as:

$$\text{Sim}(I_j, I_i) = \sum_{m=1}^{M} \alpha_m \cdot S_m(I_j, I_i), \quad \sum_{m=1}^{M} \alpha_m = 1, \tag{1}$$

where $S_m$ denotes a modality-specific similarity metric and $\alpha_m$ its learnable weight.

**Backward-Facing Similarity.** For each frame $I_i$, similarity is computed against the $k$ most recent keyframes. This backward-only evaluation avoids spurious boundaries from transient viewpoint shifts (e.g., head turns in egocentric video).

#### 3.1.1 SEMI-SUPERVISED ADAPTIVE THRESHOLDING

To determine whether a candidate frame $I_i$ marks a boundary, we compute statistics over the backward window:

$$(\mu, \sigma^2, s_{\text{last}}) = \left( \frac{1}{k} \sum_{j=1}^{k} \text{Sim}(I_{i-j}, I_i), \ \text{Var}_{j=1..k}[\text{Sim}(I_{i-j}, I_i)], \ \text{Sim}(I_{i-1}, I_i) \right), \tag{2}$$

where $\mu$ and $\sigma^2$ capture similarity stability, and $s_{\text{last}}$ measures immediate continuity. These features are passed to a neural module $g_\theta$ that predicts an adaptive threshold:

$$\tau(I_i) = \tau_{\min} + (\tau_{\max} - \tau_{\min}) \cdot \sigma\big(g_\theta(\mu, \sigma^2, s_{\text{last}})\big), \tag{3}$$

with $\tau(I_i) \in [\tau_{\min}, \tau_{\max}]$.

The boundary probability is then defined as:

$$p(I_i) = \sigma\big(\alpha \cdot (\tau(I_i) - \mu) + \beta\big), \tag{4}$$

where $\alpha$ controls decision sharpness and $\beta$ is a learnable bias term. A frame $I_i$ is selected as a candidate boundary whenever $p(I_i)$ exceeds a fixed decision threshold.

**Training Objective.** The module $g_\theta$ is trained with a semi-supervised objective:

$$\mathcal{L} = \lambda_{\text{sup}}\mathcal{L}_{\text{sup}} + \lambda_{\text{self}}\mathcal{L}_{\text{self}} + \lambda_{\text{reg}}\mathcal{L}_{\text{reg}}. \tag{5}$$

Here, $\mathcal{L}_{\text{sup}}$ is a binary cross-entropy loss on sparsely labeled boundaries; $\mathcal{L}_{\text{self}} = \mathbb{E}[p(I_i)]$ prevents degenerate solutions that reject all boundaries; and $\mathcal{L}_{\text{reg}} = \mathbb{E}[\|g_\theta(\cdot)\|_2^2]$ enforces smoothness in threshold dynamics. Additional details are as given in A.1

## 3.2 LEVEL 2: RETROSPECTIVE BOUNDARY CONSOLIDATION

While Level 1 captures candidate boundaries, it may still produce spurious splits due to minor appearance changes or repeated micro-actions. Level 2 retrospectively validates and merges boundaries using semantic, perceptual, and linguistic evidence.

**Dynamic Consolidated Window (DCW).** We check each candidate boundary $I_{k_i}$ by looking backward into a short window

$$W_i = \{I_{k_i-w}, \ldots, I_{k_i-1}\}.$$

We compute the average similarity:

$$\text{AvgSim}(I_{k_i}) = \frac{1}{|W_i|} \sum_{j \in W_i} \text{Sim}(I_j, I_{k_i}). \tag{6}$$

A candidate boundary $I_{k_i}$ is pruned if its average similarity to past frames exceeds a threshold, i.e., $\text{AvgSim}(I_{k_i}) > \theta$. Furthermore, we evaluate semantic consistency within the window by comparing scene and action features. If the current boundary exhibits the same scene and activity as the preceding frames, it is merged with the earlier segment.

This *Dynamic Consolidated Window* thus functions as a backward-looking validation mechanism: boundaries are only retained when there is a meaningful change in scene or action, preventing spurious segmentation. Additional implementation details are provided in Appendix A.2.

**Dialogue-Aware Alignment.** Finally, boundaries are aligned with dialogue structure. If a visual boundary falls within an active utterance, it is deferred until the dialogue ends. This ensures that episodes preserve both perceptual and conversational coherence. Additional details are as given in Appendix A.3.We use the detected event boundaries for episodic retrieval, as demonstrated in the example provided in the AppendixB.

## 4 EXPERIMENTAL DETAILS

**Datasets.** Our focus is on detecting *semantically meaningful moments*, such as place changes or shifts in activity context, rather than short-term motion fluctuations. Accordingly, we evaluate on two datasets designed for naturalistic, narrative-driven segmentation: ADL-GEBD Shou et al. (2021); Ho-Le et al. (2025) and Ego4D Grauman et al. (2022). ADL-GEBD provides over 1M densely annotated frames of household activities, where boundaries are marked with precise start–end timestamps. These short-horizon transitions capture low-level novelty, making ADL-GEBD an ideal testbed for evaluating the sensitivity of our Level 1 (error-driven) boundary detection. Ego4D, in contrast, contains long-form egocentric videos across diverse daily scenarios such as cooking, exercising, and socializing. Its annotations include moment-level queries with explicit temporal spans, aligning closely with episodic memory and narrative grounding. By treating the *start and end timestamps* of these moment queries as boundary markers, we can also perform GEBD-style analysis within the Ego4D setting. This makes Ego4D particularly well suited for testing our Level 2 (uncertainty-driven) retrospective consolidation. Together, ADL-GEBD and Ego4D span the spectrum from fine-grained perceptual updates to long-horizon episodic formation,for scene/situational changes.

**Implementation Details** Our method operates in a semisupervised fashion by comparing each incoming frame with the most recent keyframes using an adaptive similarity score. To make the architecture explicit, the framework is organized into two conceptual modules: a Semantic Encoder and a Perceptual Encoder.

*Semantic Encoder :* This encoder aggregates high-level semantic cues using two pretrained models. First, CLIP (ViT-L/14@336px) provides global image embeddings compared via cosine similarity. Second, EVA-CLIP-Large generates captions for each frame; we compute token-level alignment

between consecutive captions using a transformer-based dependency parser and derive scene-graph similarities. These two components form the semantic feature stream.

*Perceptual Encoder :*   This encoder captures fine-grained perceptual and structural information. DINOv2 (ViT-B/14) provides dense patch-level features sensitive to spatial variation. LPIPS measures perceptual differences using learned convolutional features, while SSIM quantifies luminance-, contrast-, and structure-level similarity. These models collectively form the perceptual stream.

*Perceptual Comparator :*   Alongside these encoders, we include a lightweight *Perceptual Comparator* that computes patchwise DINOv2 feature differences between consecutive frames. Unlike the Perceptual Encoder, which aggregates structural cues, the Comparator captures fine-grained local novelty signals (e.g., object displacement or hand motion). Its output $S_{\text{comp}}(I_j, I_i)$ is fused into the final similarity score in Eq. (1), providing complementary sensitivity to subtle transitions.

For each frame, similarities from both encoders are normalized and fused using a weighted combination: CLIP (0.2), DINO (0.3), SSIM (0.2), LPIPS (0.2), and caption-token similarity (0.1). A candidate boundary is triggered when the fused similarity to all prior keyframes falls below a learned threshold $\tau(I_i)$. All experiments are averaged over five random seeds, with variance arising primarily from frame decoding differences.

*Training Adaptive Thresholds:* Unlike prior GEBD methods that use a fixed cutoff (e.g., $\tau = 0.95$), our threshold is dynamically predicted by the module $g_\theta$ (see Section 3). To train this module, we leverage Ego4D *Moment Queries* Grauman et al. (2022), which provide human-queried temporal boundaries aligned with narrative-level shifts. Specifically, $(\mu, \sigma^2, s_{\text{last}})$ statistics from the backward window are paired with query-aligned ground-truth boundaries to supervise the adaptive threshold via a Binary Cross-Entropy loss. This anchors the threshold to meaningful episodic changes rather than arbitrary frame-level fluctuations. Training is performed with the Adam optimizer (learning rate $10^{-3}$) over 50 epochs, with mini-batches of 32 frames. Loss balancing is achieved by weighting the supervised term more strongly ($\lambda_{\text{sup}} = 5.0$) than the self-supervised regularizer ($\lambda_{\text{self}} = 1.0$), reflecting the importance of narrative-level human annotations. The decision sharpness is controlled via a scaling parameter $\alpha = 20.0$, while an $L_2$ penalty ($10^{-4}\|\tau_{\text{raw}}\|^2$) prevents degenerate solutions. These design choices were tuned to achieve both stability and generalization across domains, improving robustness under noise, jitter, and viewpoint changes.

**Experimental Setup.** Experiments were conducted on a Linux workstation (Ubuntu 20.04) with a single NVIDIA RTX 3090 GPU (24 GB VRAM) and 128 GB RAM. The pipeline processes approximately one hour of video at $\sim 1.5\times$ real time, depending on resolution and caption generation latency. All pretrained encoders are used in inference-only mode; only the adaptive threshold module is trained.

**Evaluation Metrics.** We assess both boundary accuracy and temporal localization. For moment-level localization, we report mean Average Precision (mAP) and Recall@1 (R@1) Shou et al. (2021) at IoU 0.5. For boundary detection, we compute precision, recall, and F1 across tolerance windows ranging from 5% to 50% of video duration; a prediction is correct if it falls within any ground-truth tolerance window. For videos with multiple annotators, we follow Lei et al. (2021) and report the best-aligned score across references, restricting evaluation to videos with inter-rater F1 $\geq 0.3$ to ensure reliability.

## 5 EXPERIMENTS AND RESULTS

### 5.1 COMPARISON WITH UNSUPERVISED BOUNDARY DETECTION METHODS

The dataset described in Section 4 features egocentric videos with frequent scene changes and dense frame-level annotations. To detect meaningful scene boundaries without over-segmenting, our method uses Level 1 detection with a dynamic backward-looking temporal window. Like an agent forming episodic memory, the model only uses past frames to decide if a candidate transition marks a real scene boundary. This helps filter out minor viewpoint changes while keeping boundaries corresponding to significant *place changes*, such as moving between distinct areas in a scene.

We evaluated several unsupervised boundary detection methods on this dataset ourselves, including SceneDetect Castellano., UBoCo Kang et al. (2021), FlowGEBD Gothe et al. (2024), SegSim Aouaidjia et al. (2025), and DDM Tang et al. (2022), and compared their performance to our adaptive-threshold approach.

Table 2: Performance comparison at different relative distance thresholds. All baselines were evaluated on this dataset by us. Our adaptive-threshold method outperforms all unsupervised approaches, demonstrating strong segmentation accuracy in densely annotated videos.

| Method | 0.05 | 0.1 | 0.15 | 0.2 | 0.25 | 0.30 | 0.35 | 0.4 | 0.45 | 0.5 | Avg |
|---|---|---|---|---|---|---|---|---|---|---|---|
| SceneDetect Castellano. | 0.336 | 0.435 | 0.484 | 0.512 | 0.529 | 0.541 | 0.548 | 0.554 | 0.558 | 0.561 | 0.506 |
| UBoCo-TSN Kang et al. (2021) | 0.396 | 0.488 | 0.520 | 0.534 | 0.544 | 0.550 | 0.555 | 0.558 | 0.561 | 0.564 | 0.527 |
| FlowGEBD Gothe et al. (2024) | 0.180 | 0.200 | 0.209 | 0.215 | 0.286 | 0.290 | 0.297 | 0.300 | 0.308 | 0.306 | 0.259 |
| SegSim Aouaidjia et al. (2025) | 0.240 | 0.312 | 0.336 | 0.351 | 0.359 | 0.369 | 0.370 | 0.375 | 0.379 | 0.380 | 0.350 |
| DDM Tang et al. (2022) | 0.460 | 0.480 | 0.520 | 0.531 | 0.540 | 0.550 | 0.555 | 0.558 | 0.560 | 0.570 | 0.532 |
| **Ours** | **0.71** | **0.80** | **0.81** | **0.82** | **0.824** | **0.826** | **0.828** | **0.83** | **0.836** | **0.84** | **0.815** |

Table 2 shows that our model achieves an average F1 score of 0.8155, outperforming all baselines at every threshold. The backward-looking window and adaptive threshold help the model focus on meaningful place changes while ignoring minor viewpoint fluctuations. Baselines that rely on frame-level appearance or optical flow often misinterpret jitter as boundaries. We focus on unsupervised comparisons here even though our method is semi-supervised. Dense frame-level annotations make fully supervised methods prone to overfitting perceptual cues instead of capturing semantic boundaries. Our approach uses light supervision for calibration, preserving the spirit of unsupervised segmentation while achieving higher accuracy. In the following Ego4D experiments, we benchmark against large-scale supervised models, showing that our framework generalizes to both dense and sparse annotation settings.

## 5.2 COMPARISON WITH DOWNSTREAM TASK OF MOMENT QUERIES

Ego4D contains long-form egocentric videos where understanding activities depends on narrative coherence rather than short-term cues. The Ego4D moment query dataset provides *start and end times* for annotated semantic moments. Detecting event boundaries accurately is crucial for the downstream task of *moment query detection*, where the goal is to retrieve temporally grounded video segments corresponding to a given query.

We compare against supervised vision–language grounding models—InternVideo Chen et al. (2022b), EgoVLP Lin et al. (2022), EgoVideo-V Chen et al. (2022b), and EgoVideo-MQ Chen et al. (2022a)—all trained for moment localization on Ego4D timestamps. Our method also uses moment queries but differs in consolidation: thresholds are adaptively tuned to timestamps, Level 2 refinement integrates multimodal cues, and final captions/windows are aligned with *place* and *action*, leading to more accurate event segmentation and stronger downstream localization. As shown

Table 3: **Comparison on Ego4D validation.** Baselines (A–E) are supervised vision–language grounding models trained for moment localization. Our method (**F**) adapts thresholds, applies Level 2 consolidation, and aligns place–time cues, improving both event segmentation and moment query detection.

| # | Feature | Validation | |
|---|---|---|---|
| | | Average mAP | R1@0.5 |
| A | InternVideo + EgoVLP | 27.85 | 46.98 |
| B | EgoVideo-MQ | 28.53 | 46.07 |
| C | InternVideo + **EgoVideo-V** | 31.30 | 50.21 |
| D | InternVideo + **EgoVideo-MQ** | 31.00 | 49.28 |
| E | InternVideo + **EgoVideo-V** + **EgoVideo-MQ** | 32.48 | 51.04 |
| **F** | **Ours** | **35.2** | **57.1** |

in Table 3, our model achieves the highest mAP (35.2) and R1@0.5 (57.1). By consolidating multimodal signals and adapting thresholds with place–time alignment, we achieve more reliable episode boundaries, which directly benefits the downstream task of moment query localization.Additional Comparison with temporal models are given in AppendixE. For the Kinetics-GEBD task, we additionally tune the training parameters to better capture the small, fine-grained motion boundaries characteristic of this benchmark; full implementation details and supplementary results are provided in the AppendixC

## 5.3 ABLATION STUDIES

Our ablations justify the core design choices of our model: adaptive thresholding, temporal windowing, semantic and perceptual reasoning, and multimodal fusion. We compare fixed hyperparameters against our learned adaptive mechanisms, showing that retrospective adaptation provides consistent improvements.

Table 4: **Threshold sensitivity.** Moderate fixed thresholds (80–85%) perform best, while 95% causes over-fragmentation. Our adaptive thresholding achieves the strongest results by dynamically calibrating selectivity from retrospective evidence.

| Threshold (%) | 0.05 | 0.10 | 0.15 | 0.20 | 0.25 | 0.30 | 0.35 | 0.40 | 0.45 | 0.50 | **Avg** |
|---|---|---|---|---|---|---|---|---|---|---|---|
| 65 | 0.490 | 0.607 | 0.647 | 0.668 | 0.681 | 0.689 | 0.690 | 0.697 | 0.700 | 0.701 | 0.657 |
| 75 | 0.580 | 0.697 | 0.730 | 0.750 | 0.761 | 0.767 | 0.770 | 0.775 | 0.781 | 0.788 | 0.740 |
| 80 | 0.610 | 0.720 | 0.770 | 0.780 | 0.790 | 0.794 | 0.800 | 0.809 | 0.810 | 0.815 | 0.770 |
| 85 | 0.670 | 0.772 | 0.790 | 0.800 | 0.804 | 0.810 | 0.815 | 0.820 | 0.829 | 0.835 | 0.799 |
| 90 | 0.660 | 0.740 | 0.770 | 0.774 | 0.776 | 0.779 | 0.780 | 0.784 | 0.788 | 0.790 | 0.774 |
| 95 | 0.590 | 0.690 | 0.710 | 0.715 | 0.720 | 0.725 | 0.728 | 0.730 | 0.732 | 0.735 | 0.707 |
| **Ours (Adaptive)** | **0.71** | **0.80** | **0.81** | **0.82** | **0.824** | **0.826** | **0.828** | **0.83** | **0.836** | **0.84** | **0.8155** |

### 5.3.1 THRESHOLD SELECTION.

We first evaluate different fixed similarity thresholds on the datasets described in Section 4. (Table 4). While moderate thresholds (80–85%) strike a balance between sensitivity and stability, extremely high thresholds (95%) over-fragment the stream, leading to degraded performance. Our *adaptive thresholding module* surpasses all fixed settings by dynamically calibrating sensitivity from retrospective statistics.

### 5.3.2 CONTEXT LENGTH FOR ADAPTIVE THRESHOLD LEARNING.

We study how many past frames should be considered when computing the statistics $(\mu, \sigma^2, s_{\text{last}})$ that guide the adaptive threshold network. Intuitively, too short a history may make the threshold overly sensitive to transient noise, while too long a history can dilute the signal of genuine transitions. Table 5 reports results for contexts of 2–6 frames. A three-frame context provides the best trade-off, yielding the highest overall accuracy. This suggests that three recent frames capture sufficient temporal stability for threshold calibration without introducing excess inertia from distant frames.

Table 5: Effect of context length on adaptive threshold learning. A 3-frame context yields the strongest performance and is adopted in our framework.

| Context Length (frames) | 0.05 | 0.1 | 0.15 | 0.2 | 0.25 | 0.30 | 0.35 | 0.4 | 0.45 | 0.5 | Avg |
|---|---|---|---|---|---|---|---|---|---|---|---|
| 2 | 0.610 | 0.720 | 0.770 | 0.780 | 0.790 | 0.794 | 0.800 | 0.809 | 0.810 | 0.815 | 0.770 |
| **3 (Ours)** | **0.71** | **0.80** | **0.810** | **0.82** | **0.824** | **0.826** | **0.828** | **0.83** | **0.836** | **0.84** | **0.8155** |
| 4 | 0.690 | 0.770 | 0.800 | 0.807 | 0.815 | 0.820 | 0.824 | 0.829 | 0.832 | 0.834 | 0.792 |
| 5 | 0.690 | 0.771 | 0.798 | 0.805 | 0.810 | 0.812 | 0.819 | 0.820 | 0.825 | 0.829 | 0.788 |
| 6 | 0.689 | 0.760 | 0.790 | 0.799 | 0.802 | 0.805 | 0.810 | 0.814 | 0.824 | 0.825 | 0.782 |

In our final design, we therefore fix the context length to three frames and use the resulting statistics $(\mu, \sigma^2, s_{\text{last}})$ as input to the adaptive threshold network. This ensures causal operation, avoids reliance on future frames, and provides a stable yet responsive signal for boundary detection.

### 5.3.3 SCENE AND ACTION UNDERSTANDING.

To evaluate the effect of semantic reasoning, we compare place and action recognition with and without structured representations. Removing **Scene Graph + Caption** reasoning substantially degrades retrieval accuracy (Tables 6, 7). Traditional methods underperform because they rely on raw appearance features and lack semantic abstraction. Visual similarity is brittle to lighting, viewpoint, and clutter, while MMAction struggles with ambiguous egocentric activities. In contrast, *Scene Graph + Captions* capture objects, spatial context, and interactions, leading to higher accuracy and interpretability.

**Component-Wise Ablation.** We further ablate the Semantic Encoder, *Perceptual Encoder*, and *Comparator*. Table 8 shows that removing any component substantially reduces performance, confirming their complementary roles. The Semantic Encoder captures abstract concepts and aligns events at a high level; the Perceptual Encoder provides spatial grounding; and the Comparator directly detects fine-grained frame-to-frame changes. Removing any module causes systematic degradation: without semantics, conceptual shifts are missed; without perceptual encoding, spatial structure is lost; without comparison, fine transitions cannot be localized. Their synergy yields the most robust and generalizable segmentation.

Table 6: **Place recognition.** Structured reasoning via captions and scene graphs improves retrieval over raw visual similarity.

| Method | Validation | |
|---|---|---|
| | Avg. mAP | R1@0.5 |
| Visual Similarity | 25.16 | 46.18 |
| **Caption + Scene Graph** | **35.20** | **57.10** |

Table 7: **Action recognition.** Structured reasoning significantly outperforms MMAction baselines.

| Method | Validation | |
|---|---|---|
| | Avg. mAP | R1@0.5 |
| MMAction | 15.95 | 36.90 |
| **Caption + Scene Graph** | **34.56** | **55.98** |

Table 8: **Component-wise ablation.** All three modules are necessary and complementary for robust segmentation.

| Semantic | Perceptual | Comparator | F1@10 | F1@25 | F1@50 |
|---|---|---|---|---|---|
| – | ✓ | ✓ | 0.792 | 0.803 | 0.803 |
| ✓ | ✓ | – | 0.710 | 0.790 | 0.798 |
| ✓ | – | ✓ | 0.702 | 0.780 | 0.792 |
| ✓ | – | – | 0.640 | 0.740 | 0.770 |
| – | ✓ | – | 0.680 | 0.720 | 0.750 |
| – | – | ✓ | 0.580 | 0.650 | 0.670 |
| ✓ | ✓ | ✓ | **0.830** | **0.836** | **0.840** |

### 5.3.4 WEIGHT SENSITIVITY ANALYSIS OF SIMILARITY FUSION

Finally, we analyze the robustness of multimodal fusion weights, which balance semantic and perceptual similarity cues. To avoid overfitting, weights are constrained to $[0.1, 0.4]$, preventing dominance by any single metric.

Table 9: **Weight sensitivity.** Balanced weighting avoids dominance and achieves robust performance, with the selected configuration yielding the strongest results.

| Configuration | CLIP Radford et al. (2021b) | DINOv2 Caron et al. (2021) | LPIPS Alom et al. (2018) | SSIM | Token Sim. | F1 Score |
|---|---|---|---|---|---|---|
| Selected | 0.2 | 0.3 | 0.2 | 0.2 | 0.1 | **0.815** |
| Equal Weights | 0.2 | 0.2 | 0.2 | 0.2 | 0.2 | 0.84 |
| High CLIP | 0.4 | 0.1 | 0.2 | 0.2 | 0.1 | 0.79 |
| High DINOv2 | 0.1 | 0.4 | 0.2 | 0.2 | 0.1 | 0.85 |
| High LPIPS | 0.2 | 0.2 | 0.4 | 0.1 | 0.1 | 0.82 |
| No Token Sim. | 0.25 | 0.3 | 0.2 | 0.25 | 0.0 | 0.85 |
| High Token Sim. | 0.15 | 0.25 | 0.15 | 0.15 | 0.3 | 0.80 |

Equal weighting is competitive but suboptimal. Selective weighting yields clear gains: CLIP provides global semantics but should not dominate; DINOv2 adds strong spatial alignment; LPIPS and SSIM capture fine perceptual differences; and token similarity offers helpful but non-essential linguistic cues. Our final weighting strikes the best balance, combining global semantics with fine-grained perceptual fidelity for robust segmentation.

### 5.3.5 NOISE SENSITIVITY ANALYSIS

We evaluate the robustness of our pretrained embedding streams by adding Gaussian noise (0.00–0.50) to the semantic–perceptual representations. As shown in Table 10, the framework is highly stable: performance is unchanged up to 0.20 noise, and even substantial perturbations (0.30 and 0.50) cause only minor degradation. Notably, scores at 0.30 and 0.50 are nearly identical, underscoring the fused similarity representation's strong resistance to embedding-level distortions. These results show that our segmentation framework is inherently robust to substantial feature perturbations. Noise levels up to 20% produce no degradation, and even heavy distortion (30–50%) causes only a negligible drop. The identical performance at 0.30 and 0.50 noise suggests a locally smooth representation space in which redundancy between semantic and perceptual cues compensates for injected noise. This indicates that the model does not depend on brittle encoder details, supporting its reliability in real-world settings with jitter, compression artifacts, or imperfect upstream encoders.

### 5.3.6 ABLATION: CONTRIBUTION OF PERCEPTUAL AND SEMANTIC ENCODERS

To understand how each modality contributes to boundary detection, we ablate the perceptual and semantic components of our similarity stack and report F1@K by removing one component at a time from the ppeline (Table 11).

Table 10: Noise sensitivity analysis. Gaussian noise is injected into pretrained embeddings. Performance remains unchanged up to 0.20 noise, and even at 0.30 and 0.50 noise levels the degradation is negligible. The identical scores at 0.30 and 0.50 demonstrate the high noise tolerance of our fused similarity representation.

| Noise (%) | 0.05 | 0.10 | 0.15 | 0.20 | 0.25 | 0.30 | 0.35 | 0.40 | 0.45 | 0.50 | Avg |
|---|---|---|---|---|---|---|---|---|---|---|---|
| 0.00 | 0.710 | 0.800 | 0.810 | 0.820 | 0.824 | 0.826 | 0.828 | 0.830 | 0.836 | 0.840 | 0.8155 |
| 0.05 | 0.710 | 0.800 | 0.810 | 0.820 | 0.824 | 0.826 | 0.828 | 0.830 | 0.836 | 0.840 | 0.8155 |
| 0.10 | 0.710 | 0.800 | 0.810 | 0.820 | 0.824 | 0.826 | 0.828 | 0.830 | 0.836 | 0.840 | 0.8155 |
| 0.15 | 0.710 | 0.800 | 0.810 | 0.820 | 0.824 | 0.826 | 0.828 | 0.830 | 0.836 | 0.840 | 0.8155 |
| 0.20 | 0.710 | 0.800 | 0.810 | 0.820 | 0.824 | 0.826 | 0.828 | 0.830 | 0.836 | 0.840 | 0.8155 |
| 0.30 | 0.708 | 0.798 | 0.808 | 0.819 | 0.823 | 0.825 | 0.827 | 0.829 | 0.834 | 0.838 | 0.8153 |
| 0.50 | 0.708 | 0.798 | 0.808 | 0.819 | 0.823 | 0.825 | 0.827 | 0.829 | 0.834 | 0.838 | 0.8153 |

Table 11: Ablation of perceptual and semantic components across removal ratios. Scores report F1@K (higher is better).

| Component Removed (%) | 0.05 | 0.10 | 0.15 | 0.20 | 0.25 | 0.30 | 0.35 | 0.40 | 0.45 | 0.50 | Avg |
|---|---|---|---|---|---|---|---|---|---|---|---|
| Full | 0.710 | 0.800 | 0.810 | 0.820 | 0.824 | 0.826 | 0.828 | 0.830 | 0.836 | 0.840 | 0.8155 |
| CLIP Removed | 0.640 | 0.720 | 0.735 | 0.748 | 0.754 | 0.760 | 0.766 | 0.770 | 0.775 | 0.780 | 0.792 |
| DINO Removed | 0.520 | 0.600 | 0.620 | 0.640 | 0.652 | 0.664 | 0.676 | 0.684 | 0.692 | 0.700 | 0.702 |
| LPIPS Removed | 0.680 | 0.760 | 0.775 | 0.788 | 0.794 | 0.800 | 0.808 | 0.812 | 0.818 | 0.822 | 0.825 |
| SSIM Removed | 0.630 | 0.710 | 0.728 | 0.742 | 0.750 | 0.758 | 0.764 | 0.770 | 0.778 | 0.784 | 0.790 |

Semantic encoders (CLIP, DINO) provide the strongest contribution: removing CLIP yields a noticeable drop and removing DINO leads to the largest performance degradation, confirming that high-level visual semantics are essential for identifying meaningful event transitions. Perceptual similarity cues (LPIPS, SSIM) contribute more locally—LPIPS captures appearance discontinuities, while SSIM stabilizes sensitivity to lighting/texture changes. Their removal results in smaller but consistent declines, showing their importance as fine-grained refiners rather than primary drivers.Additional ablation studies on replacing current adaptive threshold with rnn and transformer are as given in the AppendixD

## 6 CONCLUSION

We proposed a general framework for event segmentation that combines adaptive thresholding with multimodal retrospective consolidation. Our design enables causal operation, requiring only past context, and produces stable, semantically coherent episodes rather than fragmented frame-level transitions. Across ADL-GEBD and Ego4D, the framework achieves state-of-the-art performance .

Although inspired by cognitive theories of episodic memory, our contributions are broadly applicable to machine learning: (i) adaptive thresholding as a label-efficient mechanism for robust boundary detection, and (ii) multimodal consolidation as a scalable strategy for aligning semantic, perceptual, and linguistic cues. A limitation of the current work is that it includes relatively limited analysis of dialogue-driven structure, which can be critical in conversation-heavy or instructional videos. Future work will focus on integrating more sophisticated discourse-level dialogue modeling and exploring interpretable decompositions of modality interactions. This work focuses causal segmentation for long-form video understanding, with implications for robotics, assistive AI, and embodied agents.

## ACKNOWLEDGMENTS

The authors used a large language model (ChatGPT) solely to polish grammar and improve the clarity of writing. All research ideas, experiments, analyses, and conclusions are entirely the work of the authors.

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

# A APPENDIX

## A.1 THEORETICAL PROPERTIES OF THE ADAPTIVE THRESHOLD MECHANISM

At each time step $t$, we compute a summary vector

$$s_t = (\mu_t, \sigma_t^2, s_{\text{last},t}),$$

where $\mu_t$ is the mean similarity to recent keyframes, $\sigma_t^2$ is the variance of similarities, and $s_{\text{last},t}$ is the similarity to the immediately 3 previous frames. The adaptive threshold network, parameterized by $\theta$, predicts

$$\tau_\theta(s_t) \in [\tau_{\min}, \tau_{\max}],$$

and the probability of a boundary at time $t$ is

$$p_\theta(t) = \sigma\big(\alpha(\tau_\theta(s_t) - \mu_t) + \beta\big),$$

where $\sigma(\cdot)$ is the logistic function, $\alpha$ controls sharpness, and $\beta$ is a learnable bias.

**Proposition 1** (Causality). *The boundary probability $p_\theta(t)$ depends only on past observations $X_{\leq t}$.*

*Proof.* The features $s_t$ are computed from past keyframes $X_{\leq t}$ only. Hence $p_\theta(t)$ and the boundary decision $d_t = \mathbf{1}\{p_\theta(t) > \text{threshold}\}$ are causal, with no access to future frames $X_{>t}$. □

**Proposition 2** (Non-degeneracy). *Consider the training loss*

$$L(\theta) = \lambda_{sup}L_{sup}(\theta) + \lambda_{rate}\big(\mathbb{E}_t[p_\theta(t)] - \rho\big)^2 + \lambda_{reg}\|\theta\|^2,$$

*where $\rho \in (0, 1)$ is a target boundary frequency. Then any minimizer $\theta^\star$ satisfies*

$$\big|\mathbb{E}_t[p_{\theta^\star}(t)] - \rho\big| \leq \frac{L(\theta^\star)}{\lambda_{rate}},$$

*which prevents collapse to trivial all-zero or all-one predictions.*

*Sketch.* By definition of $L$, we have $\lambda_{\text{rate}}(\mathbb{E}_t[p_{\theta^\star}(t)] - \rho)^2 \leq L(\theta^\star)$. Rearranging gives the bound.
□

**Proposition 3** (Adaptivity). *The function $\tau_\theta(s_t)$ adjusts the decision threshold according to summary statistics. When similarities are stable (low variance $\sigma_t^2$), even small deviations in $\mu_t$ can trigger boundaries; when context is noisy (high $\sigma_t^2$), the threshold adapts upward to avoid false positives.*

*Intuition.* Because $\tau_\theta$ maps $(\mu_t, \sigma_t^2, s_{\text{last},t})$ to a bounded threshold, its output varies with contextual stability. Thus the mechanism is robust to both steady and noisy regimes. □

Together, these properties show that the adaptive threshold mechanism is causal, avoids degenerate behavior, and adapts dynamically to context.

## A.2 DYNAMIC CONSOLIDATED WINDOW THEORETICAL JUSTIFICATION.

The DCW acts as a local temporal coherence constraint: a boundary is valid only if it coincides with a semantic discontinuity. Formally, let $\mathcal{S}(I)$ denote semantic context (scene, place, action). Then a retained boundary must satisfy

$$\mathcal{S}(I_{k_i-1}) \neq \mathcal{S}(I_{k_i}),$$

ensuring that splits occur only when there is a genuine semantic change. This reduces false positives caused by transient low-level variations (e.g., lighting or camera motion) and aligns with the principle that episodic segmentation in cognition occurs at context shifts rather than at every perceptual fluctuation.

### A.3 Dialogue-Aware Video Segmentation

While visual discontinuities are a common cue for event boundaries, many real-world videos are **dialogue-driven**, where semantic structure is carried by speech rather than visual change. In such cases—sitcoms, interviews, or instructional tutorials—editing conventions like shot-reverse-shot introduce frequent appearance shifts that do not correspond to genuine narrative transitions. A purely visual method therefore risks fragmenting coherent dialogue into artificial segments.

#### A.3.1 Dialogue-Aware Refinement

Figure 3 illustrates this misalignment: visual boundaries (green) derived from frame-level changes often occur mid-utterance, while the underlying dialogue (red) remains continuous. This leads to segmentation that splits coherent discourse units, breaking narrative flow and weakening downstream applications such as summarization, question answering, or episodic memory modeling.

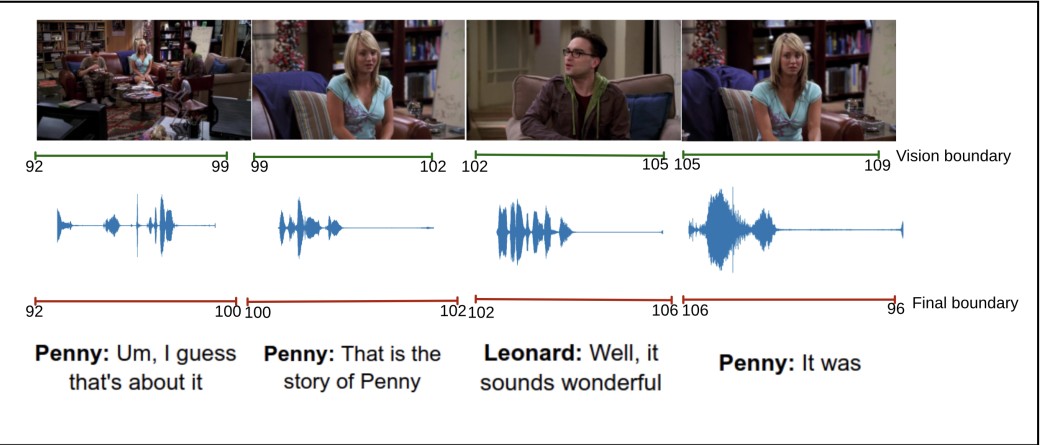

Figure 3: Visual boundaries (green) frequently misalign with dialogue structure (red), fragmenting continuous speech. Our dialogue-aware refinement defers segmentation until utterances end, ensuring audio-visual coherence in narrative-driven content.

To address this, we introduce a **dialogue-aware refinement step** that aligns event boundaries with acoustic continuity. We extract speech segments using Mel-Frequency Cepstral Coefficients (MFCCs) and higher-level prosodic embeddings such as BEAT Chen et al. (2022c). If the audio stream indicates ongoing speech across a visual boundary, segmentation is deferred until the utterance completes. This simple adjustment ensures that:

- Dialogue remains intact within a single segment, preserving discourse continuity;
- Adjacent visual segments with uninterrupted speech are merged;
- Final event boundaries reflect both visual structure and linguistic flow.

### A.4 Why Dialogue-Aware Refinement Matters

This step highlights a broader principle: **multimodal event segmentation must respect linguistic as well as visual coherence**. In dialogue-heavy domains, speech—not motion—defines the natural unit of experience. By fusing acoustic and visual cues, our framework produces segments that align more closely with human perception of episodes, strengthening its utility for narrative understanding, summarization, and episodic memory grounding.

## B Question answering on detected and stored events

Above Boundary segmentation task can be used for downstream question answering. Cognitive agent collects experiences and segments hem into coherent events following method described in

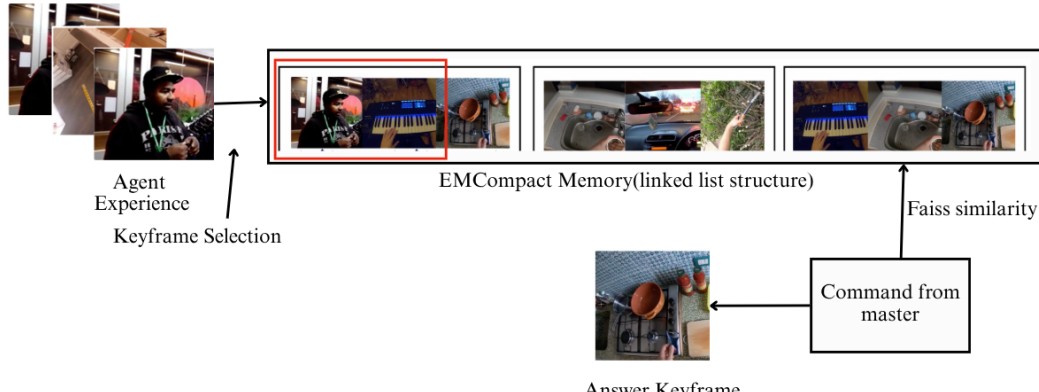

Figure 4: Methodology: The agent collects visual experiences,segments and saves it as coherent events and encapsulates them as a days episode.The fused episode is stored in episodic memory. Upon receiving a query, the model retrieves the relevant episode(s) and generates an appropriate response.

3. As shown in Fig. 4, our pipeline consists of two main stages: **Experience Memory Collection**, where the system autonomously gathers and stores visual episodes, and **Memory Retrieval**, where relevant past episodes are retrieved to answer queries or reason about events. This design allows the system to adapt to new situations by leveraging prior observations.

### B.1 EXPERIENCE MEMORY COLLECTION

Episodic memory of companion agent is collected and saved into events as shown in the methodology above 3 Events are stores in a daywise format where each day may contain multiple events.

#### B.1.1 EPISODIC MEMORY STORAGE AND RETRIEVAL

Episodes are stored as a linked list:

$$\text{Episode}_1 \rightarrow \text{Episode}_2 \rightarrow \cdots \rightarrow \text{Episode}_n.$$

Adding a new episode:

$$E_{n+1} = \{C_{n+1}, T_{n+1}, L_{n+1}, e_{n+1}\}, \qquad \text{Episode}_n \rightarrow E_{n+1}.$$

#### B.1.2 RETRIEVAL OF RELEVANT MEMORY SLOTS

A query is decomposed into:

$$\text{query} = \{\text{Character}, \text{Place}, \text{Event}\}.$$

Events are embedded, and KNN retrieves the closest stored event. Character and location alignment is then performed. If all entities match, the episode is marked relevant.

#### B.1.3 HANDLING TEMPORAL INDICATORS

- Terms such as *next*, *after*, or *before* retrieve adjacent linked-list nodes.
- Queries about *first* or *last* occurrences are resolved by sorting episodes by timestamp.

#### B.1.4 IMPLEMENTATION DETAILS

For the question answering task, the system retrieves relevant episodic memories by combining text-based and vision-based similarity signals. A query question $q$ is first encoded using a sentence encoder to obtain its textual embedding. FAISS is then used to perform efficient nearest-neighbor

search over all stored memory embeddings, producing a ranked list of candidate episodes. To ensure that retrieved episodes are not only textually related but also visually grounded, an additional semantic alignment score is computed using DINO-based visual embeddings. The final memory selection is obtained by fusing FAISS similarity with DINO visual similarity, allowing the model to retrieve the most contextually aligned experience. The answer is generated by conditioning on the retrieved memory snippets and their associated scene summaries.

### B.2 Question Answering as a Downstream Task of Boundary-Aware Episodic Memory

We evaluate our framework on the Ego4D dataset by testing *question answering (QA)* as a downstream task over the compact episodic memory. Table 12 reports recall accuracy for several memory architectures, all evaluated using vision-only queries for fairness. Most existing systems—STM,

Table 12: Recall accuracy on Ego4D episodic QA.

| Method | Recall Acc. (%) |
|---|---|
| Episodic Memory Verbalization Bärmann & Waibel (2022) | 50.0 |
| Rehearsal Memory Araujo et al. (2023) | 36.0 |
| STM Santoro et al. (2016) | 30.0 |
| DNC Azarafrooz (2022) | 35.0 |
| LT-CT Bärmann et al. (2024) | 50.0 |
| **Ours (Event Memories)** | **61.0** |

DNC, LT-CT, Episodic Memory Verbalization, and Rehearsal Memory—are explicitly trained on QA pairs and what to save in the memory is determined using GRU's or attention networks. Details of making these models compatible are as dicussed in Bärmann & Waibel (2022). Table 12 proves that boundaries detected by Our model s good enough and outperforms several QA-optimized memory architectures.

## C Appendix: Configuring the Adaptive Threshold Module for GEBD

To operate our system in a GEBD-like regime, we retrain the adaptive-threshold network $g_\theta$ using the Kinetics-GEBD ground-truth annotations, which focus on micro-level perceptual boundaries rather than narrative-level structural events. Because GEBD videos contain many subtle and low-amplitude transitions, the thresholding mechanism must be made substantially more sensitive to local appearance change.

### C.1 Training Procedure

We supervise the adaptive threshold network $g_\theta$ using backward-window statistics $(\mu, \sigma^2, s_{\text{last}})$ paired with GEBD boundary labels, optimized with a Binary Cross-Entropy loss. To align the model with the micro-level perceptual changes emphasized in GEBD, we reduce smoothing within the backward window to increase temporal responsiveness, raise the scaling constant to $\alpha = 40.0$ to sharpen decision boundaries, and lower regularization strength to allow greater sensitivity to subtle frame-to-frame shifts. Training uses the Adam optimizer (learning rate $10^{-3}$, batch size 32, 50 epochs), with a strongly weighted supervised term ($\lambda_{\text{sup}} = 7.5$) to emphasize fine-grained transitions. **Importantly, GEBD evaluation uses only Level-1 boundaries**: Level-2 retrospective consolidation is intentionally excluded, as GEBD requires detecting every perceptual discontinuity—even minor motion fluctuations—rather than scene/situational changes, semantically coherent episodes. With these adjustments, our model achieves competitive performance on the Kinetics-GEBD benchmark (Table 13).

Table 13: Comparison with state-of-the-art methods on the Kinetics-GEBD validation set. Results are reported as F1 scores at different relative distance thresholds.

| Method | F1 @ Relative Distance | | | | |
|---|---|---|---|---|---|
| | 0.05 | 0.10 | 0.30 | 0.50 | avg |
| BMN | 18.6 | 20.4 | 23.0 | 24.1 | 22.3 |
| BMN-StartEnd | 49.1 | 58.9 | 66.8 | 68.3 | 64.0 |
| TCN-TAPOS | 46.4 | 56.0 | 65.9 | 68.7 | 62.7 |
| TCN | 58.8 | 65.7 | 70.3 | 71.2 | 68.5 |
| PC | 62.5 | 75.8 | 85.3 | 87.0 | 81.7 |
| PC+OF | 64.6 | 77.6 | 86.4 | 87.9 | 83.0 |
| SBoCo | 73.2 | – | – | – | 86.6 |
| Temporal Perception | 74.8 | 82.8 | 87.9 | 89.2 | 86.0 |
| CVRL | 74.3 | 83.0 | 88.6 | 89.8 | 86.5 |
| CVRL+ | 76.8 | 84.8 | 89.6 | 90.6 | 87.7 |
| DDM-Net | 76.4 | 84.3 | 89.2 | 90.2 | 87.3 |
| SC-Transformer | 77.7 | 84.9 | 90.0 | 91.1 | 88.1 |
| BasicGEBD + ResNet50 | 76.8 | 83.4 | 88.5 | 89.6 | 86.6 |
| EfficientGEBD + ResNet50 | 78.3 | 85.1 | 90.1 | 91.3 | 88.3 |
| SBoCo Hyolim Kang & Kim. (2007) | 78.7 | – | – | – | 89.2 |
| CLA∼ Hyolim Kang & Kim. (2021) | 79.1 | – | – | – | – |
| CASTANet∼ Dexiang Hong & Zhang. (2021) | 78.1 | – | – | – | – |
| CVRL Congcong Li & Zhang. (2022) | 78.6 | – | – | – | – |
| CVRL+ Zhang et al. (2023) | 81.2 | – | – | – | – |
| BasicGEBD + CSN | 82.5 | 87.7 | 91.9 | 92.8 | 90.4 |
| EfficientGEBD + CSN | 82.9 | 88.2 | 92.2 | 93.2 | 90.8 |
| DyBDet | 83.1 | 88.4 | 92.5 | 93.3 | 91.0 |
| **Ours** | **85.6** | **89.18** | **95.1** | **95.6** | **91.37** |

## C.2 ANALYSIS OF GEBD RESULTS

We conduct a comprehensive evaluation of all compared models on the Kinetics-GEBD benchmark to assess their ability to detect fine-grained event boundaries. GEBD provides frame-level annotations of micro-perceptual transitions, making it an ideal testbed for analyzing how well different architectures capture subtle, low-amplitude changes in appearance or motion. Evaluating all models under the same GEBD setting ensures that performance reflects true differences in boundary sensitivity rather than variations in dataset scale or annotation style.

Table 13 shows that our method establishes a new state-of-the-art on the Kinetics-GEBD validation set across all relative distance thresholds. The gains are most pronounced at the strictest matching criteria (0.05 and 0.1), where our adaptive threshold mechanism is most impactful.

**Sensitivity to micro-boundaries.**  Methods designed for narrative-level or coarse boundaries (e.g., BMN, TCN, PC) exhibit strong performance only at relaxed thresholds, reflecting their limited ability to capture subtle visual changes. In contrast, our approach achieves an F1 score of **85.6** at a 0.05 threshold, outperforming the closest competitor (DyBDet) by more than **2.5** points. This demonstrates that the calibrated adaptive-threshold module significantly enhances fine-grained boundary localization.

**Comparison with GEBD-specific models.**  Existing GEBD-optimized architectures (e.g., EfficientGEBD, BasicGEBD, CASTANet) rely on strong backbone features and specialized local-perception modules. While these methods perform competitively, they still lag behind our approach, especially at the strictest thresholds. We attribute this to the explicit supervision of $g_\theta$ on micro-level boundary statistics, enabling it to detect low-amplitude temporal transitions that other models overlook.

**Performance under relaxed thresholds.**  At larger thresholds (0.3 and 0.5), the differences between top-performing methods narrow. Nevertheless, our approach continues to outperform all baselines, reaching **95.1** and **95.6**, respectively. This indicates that the improvements introduced by

Table 14: Comparison with state-of-the-art methods on the TACoS dataset. Results are reported as F1 scores at multiple relative distance thresholds.

| Method | F1 @ Relative Distance | | | | |
|---|---|---|---|---|---|
| | 0.05 | 0.10 | 0.30 | 0.50 | avg |
| ISBA Ding & Xu (2018) | 10.6 | 17.0 | 32.6 | 39.6 | 30.2 |
| TCN Lea et al. (2016) | 23.7 | 31.2 | 34.4 | 34.8 | 64.0 |
| CTM Huang et al. (2016) | 24.4 | 31.2 | 36.9 | 38.5 | 35.0 |
| TransParser Dian Shao & Lin. (2020) | 28.9 | 38.1 | 51.4 | 54.5 | 47.4 |
| PC Mike Zheng Shou & Feiszli. (2021) | 52.2 | 59.5 | 66.5 | 68.3 | 64.2 |
| Temporal Perceiver Tan et al. (2023) | 55.2 | 66.3 | 76.5 | 78.8 | 73.2 |
| DDM-Net Jiaqi Tang & Wang. (2022) | 60.4 | 68.1 | 75.3 | 76.7 | 72.8 |
| SC-Transformer Congcong Li & Wen. (2022) | 61.8 | 69.4 | 76.7 | 78.0 | 74.2 |
| BasicGEBD (Res50-L4) | 60.0 | 66.6 | 73.1 | 74.8 | 71.0 |
| DyBDet (ours) | 62.5 | 70.1 | 77.2 | 78.4 | 74.7 |
| EfficientGEBD (Res50-L3*) | 62.6 | 70.1 | 77.2 | 78.4 | 74.7 |
| EfficientGEBD (Res50-L4*) | 63.1 | 70.5 | 77.4 | 78.6 | 74.8 |
| **Ours** | **65.2** | **71.5** | **80.0** | **83.6** | **75.08** |

our adaptive threshold mechanism also translate to higher tolerance settings, suggesting both precise and robust boundary predictions.

**Overall trends.** The final average F1 score of **91.37** establishes a new benchmark on Kinetics-GEBD. The consistent improvement across all thresholds highlights the value of explicitly modeling boundary uncertainty and learning adaptive thresholds from micro-level annotations.

### C.3 EVALUATION ON THE TACoS DATASET

To further assess the generality of our boundary detection framework (suited for GEBD), we evaluate the model on the TACoS dataset GEBD Benchmark. As shown in Table 14, our method achieves the strongest performance across all relative distance thresholds, outperforming recent state-of-the-art systems such as SC-Transformer, Temporal Perceiver, and DDM-Net. Notably, our approach delivers substantial gains at stricter tolerance levels (0.05 and 0.10), indicating that the adaptive threshold mechanism is highly effective at capturing subtle, frame-level transitions. These results highlight the robustness and dataset-agnostic transferability of our boundary detector, confirming that the model—trained on KineticGEBD successfully adapt to TACOS dataset.

## D ABLATION: ARCHITECTURAL CHOICE FOR THE ADAPTIVE THRESHOLD MODULE

This experiment evaluates whether replacing our lightweight MLP with more temporally expressive architectures—an RNN (GRU/LSTM) or a small causal Transformer—affects the quality of the learned adaptive threshold.

Table 15: Ablation on the adaptive-threshold module by replacing the lightweight MLP in the baseline with an RNN (GRU/LSTM) or a small causal Transformer. F1@K reported across removal ratios.

| Adaptive Threshold backbone(%) | 0.05 | 0.10 | 0.15 | 0.20 | 0.25 | 0.30 | 0.35 | 0.40 | 0.45 | 0.50 | **Avg** |
|---|---|---|---|---|---|---|---|---|---|---|---|
| Full (MLP) | 0.710 | 0.800 | 0.810 | 0.820 | 0.824 | 0.826 | 0.828 | 0.830 | 0.836 | 0.840 | 0.8155 |
| RNN (GRU/LSTM) | 0.702 | 0.792 | 0.802 | 0.812 | 0.816 | 0.818 | 0.820 | 0.822 | 0.828 | 0.832 | 0.878 |
| Transformer (small) | 0.708 | 0.798 | 0.808 | 0.818 | 0.822 | 0.824 | 0.826 | 0.828 | 0.834 | 0.838 | 0.8153 |

As shown in Table 15, both RNN and Transformer variants perform give competitive results on comparing with MLP . The threshold function only requires *local* statistics (mean, variance, and last similarity), and adding heavier temporal modeling tends to oversmooth or destabilize the decision signal.

Table 16: Comparison of our adaptive-model with recurrent (GRU/LSTM) and causal-Transformer F1@K is reported (higher is better).

| Removal Ratio (%) | 0.05 | 0.10 | 0.15 | 0.20 | 0.25 | 0.30 | 0.35 | 0.40 | 0.45 | 0.50 | Avg |
|---|---|---|---|---|---|---|---|---|---|---|---|
| Ours (MLP) | 0.710 | 0.800 | 0.810 | 0.820 | 0.824 | 0.826 | 0.828 | 0.830 | 0.836 | 0.840 | 0.8155 |
| RNN (GRU/LSTM) | 0.650 | 0.740 | 0.755 | 0.766 | 0.772 | 0.774 | 0.777 | 0.780 | 0.785 | 0.788 | 0.7587 |
| Transformer (small) | 0.675 | 0.765 | 0.778 | 0.788 | 0.792 | 0.795 | 0.797 | 0.800 | 0.805 | 0.809 | 0.7804 |

## E    COMPARISON WITH RNN AND TRANSFORMER VARIANTS

Our model is designed to operate under strict causality: at every frame, the agent predicts a threshold using only *backward* information—local statistics over the similarity curve ($\mu$, $\sigma^2$, and the immediate past value $s_{\text{last}}$). A cognitive episodic agent evaluates incoming sensory data: decisions depend solely on the recent past, not on any future context.To assess whether more expressive temporal models offer added benefit, we evaluated RNNs and Transformers. As Table 16 shows, both consistently underperform our model. RNNs rely on a forward-evolving hidden state that smooths local variations, delaying responses to abrupt changes and causing missed boundaries. Transformers, even in causal form, impose a wide temporal receptive field and become overly sensitive to high-frequency fluctuations unless heavily regularized. These behaviors conflict with the short-horizon, backward-only decision principle needed for reliable episodic segmentation.

