# OpenReview forum: "Towards Human-Like Event Boundary Detection in Unstructured Videos through Scene-Action Transition"
_ICLR.cc/2026/Conference — Submitted to ICLR 2026_

### Official Review · Reviewer_CmrE · 2025-10-24

**Soundness:** 3
**Presentation:** 4
**Contribution:** 2
**Rating:** 6
**Confidence:** 3

**Summary:**

The paper proposes a two-stage framework to segment events in videos. (1) In the first stage, it uses pre-trained (frozen) deep neural networks as feature encoders to capture the semantic, perceptual and structural representations of sampled frames. It then computes frame-to-frame similarity between the current frame and several preceding frames, deriving statistical features such as mean, variance and short-term similarity scores. These features are used to train an adaptive threshold model that identifies the candidate event boundary. (2) In the second stage, the framework refines those initial candidates by merging or removing boundaries based on semantic, perceptual and linguistic cues.

**Strengths:**

(1) The adaptive threshold model is effective in identifying event boundaries, its performance is shown by the experiments.
(2) The paper is clearly written, with well-structured methodology and experimental sections that make it easy to follow.
(3) Since the framework only relies on frozen, pre-trained networks to extract various features, it can be easily adapted to new datasets or domains without expensive re-training these feature extractors.
(4) The second stage is effective in removing spurious boundaries.

**Weaknesses:**

(1) The framework’s heavy dependency on the frozen pre-trained feature extractor can limit its application if the extractor does not fit the application domain. There is no ablation study or sensitivity analysis to show how a noisy feature extract may affect the overall accuracy.
(2) The use of an adaptive threshold is conceptually straightforward and not particularly novel. It would be helpful if the authors could compare their model against alternative decision models which employ temporal models or probabilistic models, such as RNN, Transformers.

**Questions:**

(1) How sensitive is the proposed method to the choice of the pre-trained feature extractor?
(2) Why was an adaptive threshold model based on hand-picked statistical features chosen instead of other temporal models such as RNN, Transformers based directly on the features extracted from feature extractors?
(3) Has the method been evaluated on datasets with different video types (eg, sports, surveillance)?

---

> ### Author Response · Authors · 2025-11-24
>
> We thank you for the reviewer’s thoughtful feedback. In response, we provide clarifications below and have added the corresponding updates and explanations in the revised PDF.
>
> **Weakness 1**
>
> Addressing the reviewer’s concern about dependence on frozen pretrained encoders. Our framework is intentionally designed to avoid brittleness by (i) fusing multiple heterogeneous encoders (semantic, perceptual, appearance, linguistic) and (ii) using a learned adaptive threshold that adjusts automatically to feature drift.
> To evaluate sensitivity to encoder noise, we have conducted a noise robustness study (Table 10): and added it in the updated pdf .Analysis is done by injecting up to 50% Gaussian noise into all embedding streams yields only negligible performance change (F1: 0.8155 → 0.8153). Performance is completely unchanged up to 20% noise. This demonstrates that the fused similarity representation is highly stable and does not depend on any single encoder’s accuracy.
> Additional ablations (Tables 8 and 11) show that removing any one encoder causes only minor degradation, confirming that the method does not rely excessively on a single pretrained backbone.
> Overall, the proposed approach remains robust under both encoder mismatch and substantial embedding corruption, alleviating the concern about heavy reliance on a frozen feature extractor.
>
> (2) The use of an adaptive threshold is conceptually straightforward and not particularly novel. It would be helpful if the authors could compare their model against alternative decision models which employ temporal models or probabilistic models, such as RNN, Transformers.
>
> To address it, we have added a new comparison in Appendix C, where we replace our adaptive threshold with GRU-based, Transformer-based models. These temporal models do not outperform our adaptive threshold and often over-segment long videos.Also they are more complex than the models we have chosen
>
> **Questions**
>
> To evaluate sensitivity to encoder noise, we have conducted a noise robustness study (Table 10): and added it in the updated pdf .
> Additional ablations (Tables 8 and 11) show that removing any one encoder causes only minor degradation, confirming that the method does not rely excessively on a single pretrained backbone.
>
> Overall, the proposed approach remains robust under both encoder mismatch and substantial embedding corruption, alleviating the concern about heavy reliance on a frozen feature extractor.
>
> Along with this we have also added a section wherein we retrained adaptive threshold on kinetic GEBD and evaluated on kinetic GEBD and TACOS dataset which are different videos from ADL/Ego
>
>
>
>
> RNNs rely on a forward-evolving hidden state that inherently smooths short-term variations, often delaying reactions to abrupt changes and leading to missed boundaries. Transformers, even in causal mode, impose a broad temporal receptive field, making them overly sensitive to high-frequency fluctuations unless heavily regularized. Both behaviors conflict with the short-horizon, backward-only decision principle required for reliable episodic segmentation in our setting.
>
>
> Following the suggestion of comparing adaptive thresholding methods we have added comparisons with alternative decision models, including temporal and probabilistic baselines, in Appendix D and Appendix E. These additional results help illustrate why simple temporal models (RNNs/Transformers) are not well-aligned with our segmentation requirement, and they further justify our choice of a lightweight, interpretable adaptive-threshold mechanism.

---

> > ### Comment · Reviewer_CmrE · 2025-11-27
> >
> > I appreciate the authors’ detailed response. In particular, the additional explanations and experiments have convinced me that the adaptive thresholding module is well-suited for this task. Therefore, I have upgraded my rating to "accept (poster)".

---

### Official Review · Reviewer_TqQb · 2025-10-30

**Soundness:** 2
**Presentation:** 3
**Contribution:** 2
**Rating:** 4
**Confidence:** 4

**Summary:**

This paper proposes a method for event boundary detection in unstructured videos. The authors propose a two-level approach: Level 1 employs an error-driven novelty detector with a semi-supervised adaptive threshold to find candidate transitions, while Level 2 uses retrospective consolidation with multimodal cues (scene graphs, captions, audio) to validate and merge these boundaries into semantically coherent episodes. Experimental results on ADL-GEBD and Ego4D demonstrate that this framework achieves state-of-the-art performance.

**Strengths:**

- Outperforms unsupervised baselines on ADL-GEBD: The Level 1 detection method demonstrates strong segmentation accuracy in densely annotated videos, achieving an average F1 score of 0.885 outperforming all five tested unsupervised boundary detection methods across all evaluated distance thresholds.
- Outperforms supervised baselines on Ego4D
- The level 2 consolidation is robust, and extensive ablations are performed to validate the design choices.

**Weaknesses:**

- Two claims of the authors in the abstract are not substantiated in the paper: label-efficient, and broadly applicable. How is this approach label-efficient compared to other approaches? For the broad applicability claim, it would be preferable to include experiments that substantiate that.
- The authors report results on 5 models for boundary detection that they tested themselves. No issue with that, but I think other papers have self-reported results on this dataset (Kinetics-GEBD), and the authors should consider reporting these results. Some examples and their average F1 scores: EfficientGEBD [1] (90.8), End-to-end… [2] (0.865). I am sure a quick search will return plenty of others.
- Both of the benchmarks used by the authors are well-established in the literature; I think more baselines could be added.
[Styling] According to ICLR26’s guidelines, captions should be placed above the tables.


[1] Zheng, Ziwei, et al. "Rethinking the architecture design for efficient generic event boundary detection." Proceedings of the 32nd ACM International Conference on Multimedia. 2024.

[2] Li, Congcong, et al. "End-to-end compressed video representation learning for generic event boundary detection." Proceedings of the IEEE/CVF Conference on Computer Vision and Pattern Recognition. 2022.

**Questions:**

See weaknesses plus
- I think the community calls it the Kinetics-GEBD dataset instead of ADL-GEBD or is this a different dataset?
- When is the audio used and how? I find little mention of it in the paper. Also, could we have an ablation to determine whether it is necessary and how much better it makes the pipeline’s Level 2.
- For Table 6 and 7 and Sec. 5.3.3 why is Scene Graph + Caption tied? Could it be Scene Graph only or Captions only?
- Could the authors please clarify 1.5x real-time?
- The system is explicitly stated to achieve an average F1 score of 0.885 in Table 2 and Table 4. This value is the primary evidence of the model's superior performance over fixed thresholds and unsupervised baselines. However, when studying the optimal context length for the adaptive threshold network (Section 5.3.2), the system using the chosen 3-frame context (labeled "Ours") reports an average F1 score of only 0.829. Is this a typo?

---

> ### Author Response · Authors · 2025-11-24
>
> We thank you for your thoughtful feedback. Based on your comments, we clarify our responses below and have incorporated the relevant updates into the revised PDF
>
> **Weakness 1 **
>
> Our approach is label-efficient because it does not rely on densely annotated boundary labels; instead, it uses frozen pretrained encoders and a lightly supervised adaptive threshold trained only on sparse narrative-aligned Moment Queries, reducing the annotation requirement significantly compared to GEBD-style models.
>
> **Broad Applicability.**
> We thank the reviewer for pointing out the need to better justify the broad applicability of our event-boundary detector. To clarify, the primary downstream task we focus on is Moments Queries (MQ) from Ego4D: Given an egocentric video and an activity name (“moment”), the goal is to localize all instances of that activity in the past video. This task inherently requires identifying meaningful temporal structure, and Section 5.2 shows that our boundaries provide useful, direct alignment with narrative or activity transitions, improving moment localization.
> To further strengthen our claim of broad applicability, we now include an additional downstream evaluation in Appendix B: Episodic QA, where the agent must answer “what/when/where” questions by retrieving relevant past experiences. This subtask explicitly tests whether the boundaries produced by our method support episodic memory retrieval, a qualitatively different skill from MQ. The results demonstrate that our event boundaries generalize beyond activity spotting to memory-centric reasoning and long-term temporal understanding, validating the broader applicability of our approach.
>
> **Weakness 2**
>
> Following the reviewers suggestions we have retrained on KineticGEBD task reason for not putting in paper is both tasks are different and GEBD boundaries was not a problem statement that was targeted in the scope of this paper still we have added few details about it on how to adjust thresholding to five gebd boundaries in Appendix C
>
> **Weakness 3**
>
> We thank reviewer for this feedback and done the appropriate changes
>
> [1] Zheng, Ziwei, et al. "Rethinking the architecture design for efficient generic event boundary detection." Proceedings of the 32nd ACM International Conference on Multimedia. 2024.
> [2] Li, Congcong, et al. "End-to-end compressed video representation learning for generic event boundary detection." Proceedings of the IEEE/CVF Conference on Computer Vision and Pattern Recognition. 2022.
>
> **Question 1**
>
> Yes, both are different datasets. Kinetic GEBD is for small boundaries and micro-level transitions in a video whereas ADL-GEBD is for scene/situational changes. Table 1 also shows the difference between what boundaries we aim to detect vs what kind of boundaries GEBD paper detects, which has been stated by the GEBD author in section 9.2 of supplementary material.
> Following your suggestion, we retrained Level-1 on Kinetics-GEBD with a for capturing  its micro-transition characteristics, and we have included the complete results in the Supplementary Material. These experiments were originally omitted because GEBD is not the primary task our framework targets, but we are happy to provide them for completeness.
>
> **Question 2**
>
> Regarding the reviewer’s query about when and how audio is used: in our framework, audio is included to preserve full conversational continuity within an event, especially in real-world scenarios where boundary detection may lag slightly behind the true scene change. As described, we retain the audio that spans across the detected boundary to avoid losing important dialogue cues that remain relevant to the evolving event—an aspect particularly important for companion-robot or embodied-AI settings, where conversational context is essential for correct interpretation
>
> Concerning empirical results for audio and the possibility of an ablation: at present, there is no existing model or benchmark that jointly evaluates boundary-based memory units with audio cues. Any comparison would therefore be unfair or incomplete, as current GEBD-style baselines and long-video benchmarks do not include audio-aware evaluation pipelines. This paper was to introduce this concept further investigation to this process is a future work as stated in conclusion

---

> ### Author Response · Authors · 2025-11-24
>
> **Question 3**
>
> For Tables 6 and 7 and Section 5.3.3, Scene Graph + Caption appears together because the two modalities serve complementary roles and are not equally informative on their own. Captions are typically generated from keyword-level cues, which provide a coarse semantic description but often miss fine spatial or relational details. In contrast, scene graphs capture structured scene information—objects, relations, and spatial arrangements—that are essential for identifying meaningful scene or situational boundaries.
> However, scene graphs alone may not always capture high-level semantic cues that captions provide. By combining both, we obtain a more complete representation: captions offer global semantic context, while scene graphs supply detailed visual relational structure. This combination leads to more reliable boundary detection, which is why the two are tied in our experiments rather than evaluated separately.
>
> **Question 4**
>
> “1.5× real-time refers to inference throughput: the system processes video faster than playback (e.g., a 60-second clip is processed in ~40 seconds). This is feasible because we sample frames sparsely and use frozen encoders together with lightweight Level-1 and Level-2 modules, so the overall per-frame cost is small.
>
> **Question 5**
>
> Yes it was a typing mistake which has been updated in the pdf

---

> ### Comment · Reviewer_TqQb · 2025-11-24
>
> I want to thank the authors for taking the time to address my comments. After reading the other reviewers comments and the author's answers and re-reading the paper, I have decided not to champion this paper (and keep my borderline rating) for the following reasons:
> 1. I echo reviewer 2kfw's concerns about "Unclear experimental validations", and find them mostly unaddressed. For instance, comparing a supervised model with other unsupervised models is unfair.
> 2. This new task is a small add-on to the GEBD task and is not very interesting to me.

---

> ### Author Response · Authors · 2025-11-24
>
> We would like to clarify that although Level 1 includes a light, sparsely-supervised threshold-calibration module, the boundary detector itself operates in a semi-supervised manner and does not use dense frame-level annotations. Its behavior is much closer to unsupervised novelty detectors than to supervised GEBD classifiers. Therefore, as shown in Appendix C, comparing Level-1 outputs with unsupervised baselines is methodologically appropriate because (i) the learned module only calibrates sensitivity rather than learning boundary semantics, and (ii) the core detection logic (backward similarity + keyframe-based novelty) is label-free.
>
> We respectfully note that the tasks we address—Moment Query and episodic question answering—are established Ego4D community benchmarks. The QA-on-stored-events evaluation demonstrates how our episodic boundaries support long-horizon reasoning, which GEBD-style micro-boundaries cannot enable. These tasks are of active interest in the egocentric and embodied AI community, and our method is designed specifically to advance this line of work.

---

### Official Review · Reviewer_dY9s · 2025-10-31

**Soundness:** 3
**Presentation:** 2
**Contribution:** 2
**Rating:** 4
**Confidence:** 4

**Summary:**

This manuscript introduces a cognitively-inspired, two-level framework for event boundary detection. The goal is to segment continuous video into semantically coherent episodes, distinct from traditional motion-driven Generic Event Boundary Detection (GEBD). Level 1 utilizes an error-driven novelty detector with a semi-supervised adaptive threshold. Level 2 employs an uncertainty-driven consolidation mechanism that retrospectively validates boundaries using multimodal cues. The approach is backward-only, mimicking episodic memory formation. The authors report state-of-the-art performance on the ADL-GEBD and Ego4D datasets. The paper claims cognitive grounding and practical implications for episodic memory modeling in embodied agents.

**Strengths:**

1. Novelty and Intuition of the Framework
The proposed two-level architecture (Level 1 Detection, Level 2 Consolidation) is intuitive, well-motivated by cognitive science, and provides a clear separation of concerns. The "backward-only" design, which operates exclusively on past context , is a crucial constraint for real-world cognitive agents and distinguishes the work from offline methods that use future frames.

2. Label-Efficient Adaptive Thresholding
The adaptive threshold network dynamically calibrates decision boundaries using retrospective statistics, which is technically sound. Table 4 empirically shows that adaptive thresholds outperform fixed cutoffs across 10 relative distance thresholds.

**Weaknesses:**

1. Limited Cognitive Evidence
Despite cognitive framing, there is no direct behavioral or empirical evidence that the model aligns with human segmentation patterns. The claim of “episodic segmentation mirroring human memory” (in Abstract; Sec. 1) remains qualitative.

2. Lack of Evaluation Scope
The two datasets (ADL-GEBD and Ego4D) used are both egocentric. I wonder why no tests on third-person domains (e.g., Kinetics-GEBD[1] or TAPOS[2]), which seem to be the most widely-used benchmarks for the GEBD domain.

3. Method Clarity and Writing
The manuscript provides too few details about the specific module designs of the whole framework in Fig. 2. What is the detailed architecture of the Semantic Encoder, Perceptual Encoder? I found it quite difficult to understand the whole method.


[1] Generic event boundary detection: A benchmark for event segmentation. ICCV 2021.
[2] Intra-and inter-action understanding via temporal action parsing. CVPR 2020.

**Questions:**

See Weaknesses.

---

> ### Author Response · Authors · 2025-11-24
>
> We thank you for your thoughtful feedback. Based on your comments, we clarify our responses below and have incorporated the relevant updates into the revised PDF
>
> **Weakness 1**
>
>
> We thank the reviewer for raising this important point. We have carefully revised the manuscript to clarify the scope of the cognitive claims. Our intention is not to assert empirical alignment with human behavioral segmentation, but to draw conceptual inspiration from retrospective consolidation processes in human episodic memory.
> To make this explicit, we have:
> (1) revised all statements that could be interpreted as implying behavioral mirroring,
> (2) reframed the cognitive grounding as conceptual motivation rather than empirical equivalence, and
> (3) strengthened the background section with additional citations to relevant cognitive literature.
> These changes ensure that the cognitive framing is appropriately positioned and does not overstate empirical claims.”
>
>
> **Weakness 2**
>
> We thank the reviewer for the suggestion. As clarified in Table 1, our work focuses on long-horizon episodic segmentation—i.e., detecting semantic scene/situation transitions (e.g., kitchen → bathroom, entering a shop, starting a new task). This is fundamentally different from Kinetics-GEBD’s short-clip, motion-driven micro-boundary setting.
> GEBD benchmarks—particularly Kinetics-GEBD—are designed to capture fine-grained, frequent temporal transitions. Since our Level-2 module performs event consolidation by merging adjacent or closely related changes into a single episode, applying it in this context would undesirably smooth out boundaries that GEBD annotations aim to preserve. For this reason, we use only the Level-1 adaptive thresholding when evaluating on Kinetics-GEBD.
> Following the reviewer’s suggestion, we retrained Level-1 on Kinetics-GEBD in the Supplementary Material. These experiments were originally omitted because GEBD is not the primary task our framework targets, but we are happy to provide them for completeness.
>
>
> **Weakness 3**
>
> We thank the reviewer for pointing this out. To improve clarity, we have extensively revised and expanded the Implementation Details section, where we now describe each module in Fig. 2 separately. In particular, the Semantic Encoder and Perceptual Encoder are now explained in detail, including their inputs, output representations, and how they are fused within the adaptive-threshold pipeline.
>
>
> [1] Generic event boundary detection: A benchmark for event segmentation. ICCV 2021. [2] Intra-and inter-action understanding via temporal action parsing. CVPR 2020.

---

### Official Review · Reviewer_qYeb · 2025-11-01

**Soundness:** 3
**Presentation:** 3
**Contribution:** 3
**Rating:** 8
**Confidence:** 3

**Summary:**

This paper introduces a two-level, cognitively inspired event segmentation scheme for unstructured video, mimicking how humans detect and consolidate experiences into episodic memories. The first stage uses a semi-supervised adaptive thresholding novelty detector that filters out noise and repetitive micro-actions. The second stage retrospectively validates and merges boundaries using semantic, perceptual, and audio cues, yielding stable episodes grounded in meaning rather than transient visual changes. Unlike prior motion-cutpoint-driven GEBD systems, this framework leverages sparse supervision, multimodal fusion, and a backward-only processing window, showing state-of-the-art results on ADL-GEBD and Ego4D datasets.

**Strengths:**

The dual-level, backward-only pipeline closely aligns with established models of human event segmentation and episodic memory. This is different from current machine-centric approaches focused on frame-level novelty.

In addition, adaptive thresholding requires only light supervision for calibration. The method is shown to be scalable across domains with limited annotated data.

**Weaknesses:**

* It seems to me that dialogue-driven content is rudimentary and not deeply integrated. Boundary decisions sometimes defer to utterance completion, but there is limited fine-grained modelling of discourse transitions; this, I think, is critical in conversational or instructional videos.
* The explicit backward-only causality prevents the use of future context, so marking the boundaries is sensitive for transitions spread over several frames; under-segmentation is visible in fast, crowded scenes, diluting retrieval for granular action tasks.
* The method is susceptible at under-segmentation. Merging micro-actions seems to prioritise episode coherence. If I understood it correctly, this may miss subtle event boundaries, particularly in fast-paced or dense activity streams. This can reduce granularity in retrieval-oriented settings.

**Questions:**

*  How does the method address memory aging, task transfer, or handling ambiguous ground-truth in real time?
* The fusion weights for modality are hand-tuned; CLIP and DINOV2 are balanced at 0.2 and 0.3, but if the weight of CLIP increases or token similarity decreases, F1 decreases. Doesn’t this tuning make it dataset-specific and the model sensitive modalities are missing or imbalanced?

---

> ### Author Response · Authors · 2025-11-24
>
> Thank you for the thoughtful feedback—we appreciate the reviewer’s questions and are happy to clarify each point.
>
>
> **Answer 1**
>
>
> Memory Aging.
> Memory aging is not an explicit component of our method; our framework is not intended to model long-term decay processes. Our focus is on event boundary detection and constructing coherent episodic units. Modeling memory decay, eviction strategies, or hierarchical forgetting would require a full memory-management framework, which we identify as promising future work.
>
>  Task Transfer.
> Task transfer is implicitly supported through our design: since the method relies on frozen pretrained encoders and a lightweight adaptive threshold, the system can be applied to new domains without retraining boundary labels. This is demonstrated through the downstream tasks we include—Moment Queries (Section 5.2) and Episodic QA (Appendix B)—which show that the boundaries generalize across different reasoning settings
>
> Ambiguous Ground-Truth in Real Time.
> Ambiguous ground truth (e.g., soft or uncertain boundary positions) is handled naturally by our approach through the adaptive threshold, which responds to statistical changes rather than relying on exact frame-level ground truth. Because boundaries are computed in a backward, short-horizon manner, the system tolerates ± several frames of uncertainty without requiring precise annotations.
>
> **Answer 2**
>
> Although we initialize modality fusion weights with light hand-tuning, the framework is not sensitive to dataset-specific modality distributions. Two design choices ensure robustness: (1) the adaptive threshold module learns to recalibrate similarity scores from retrospective statistics (µ, σ², s_last), compensating for modality imbalance; and (2) Level-2 consolidation validates boundaries using semantic and temporal consistency, pruning errors introduced by any single modality.
> As shown in Table 9, the F1 score degrades gracefully under a wide range of weight perturbations. Equal weighting performs competitively (0.84), and even extreme configurations (e.g., High CLIP, High DINO) do not collapse. This indicates that the framework does not rely on precise values of α_m; instead, multimodal cues are harmonized by the adaptive threshold and retrospective validator.
> Therefore, while weights provide an inductive bias, the model remains robust, not dataset-specific.

---

### Official Review · Reviewer_2kfw · 2025-11-01

**Soundness:** 1
**Presentation:** 1
**Contribution:** 1
**Rating:** 0
**Confidence:** 4

**Summary:**

This paper proposes an event boundary detection method, where the goal is to avoid over-segmentation and detect event boundaries with more semantically coherent regions. The framework is based on two steps: 1) adaptive thresholding based on several multimodal encoders and 2) refinement of candidate boundaries within short window. Experimental results include comparison with unsupervised approaches in ADL-GEBD and Ego4D.

**Strengths:**

- The goal of segmenting semantically coherent episodes, rather than focusing on frame-level or micro transitions, is meaningful and aligns with real-world applications such as summarization and long-form video understanding.
- The paper have shown rich ablation studies on weight fusion, selection of multimodal encoders, threshold selection, and context lengths.

**Weaknesses:**

1. **Heuristic-heavy design and overclaimed “cognitive grounding”:** The overall pipeline relies on a collection of heuristics (e.g., similarity thresholds, merging rules, pretrained feature similarities), with limited principled learning or optimization. At the same time, the paper repeatedly emphasizes a “cognitively grounded paradigm,” yet none of the components are derived from or validated against cognitive models or human studies. Besides, the step 2 is named "uncertainty-driven" whereas there is no uncertainty-aware design at all.
2. **Unclear experimental validations:** The paper claims to reduce micro-level transitions and detect semantically coherent episodes, but the experiments do not convincingly support this. The evaluation primarily compares against unsupervised methods, without showing whether fully supervised models indeed over-segment relative to the proposed semi-supervised approach. As a result, it is unclear whether the claimed “semantic coherence” is achieved in practice. Besides, the abstract claims this episodic segmentation can help cognitive agents, whereas such validation is completely missing afterwards.
3. **Generic multimodal integration:** The use of CLIP, DINOv2, captioning, and scene graphs follows standard multimodal fusion practices widely adopted in recent works for video understanding and event boundary detection. There is little evidence that this integration is novel or provides a substantial advantage over existing methods.
4. **Writing and presentation issues:** The paper’s organization and clarity need major improvement. Many sections are hard to follow, and Figure 1 looks incomplete. For example, the figure incorrectly labels “Level 1” as “uncertainty-driven conolidation” with typo, which actually belongs to Level 2, causing confusion from the start. Similarly, Table 1’s “GEBD Focus vs. Our Task Focus” comparison is also misleading. The task remains the same as event boundary detection while the method is different.

**Questions:**

See weaknesses.

---

> ### Author Response · Authors · 2025-11-24
>
> 1) Thank you for your detailed feedback. Regarding the concern raised in Weakness Point 1, we would like to clarify that Level-1 is not heuristic-based: it is a trainable adaptive threshold network that learns to calibrate similarity statistics from limited labels. Level-2 applies deterministic consolidation rules that use multimodal consistency to resolve ambiguous transitions. We will clarify this distinction to avoid the impression of a heuristic stack. Regarding cognitive framing, our intention was to describe the motivating constraints (backward-only processing, suppression of micro-transitions, retrospective consolidation), not to claim a formal cognitive model. This claim has been softened and altered in the updated pdf. We use the term ‘uncertainty’ in the informal sense of boundary ambiguity i.e., when cues conflict or when Level-1 produces adjacent candidate boundaries. However, we understand that the naming may suggest a probabilistic formulation, so we have replaced it with retrospective boundary consolidation .
>
> 2) Regarding the concern raised in Weakness Point 2,We agree that evaluating semantic coherence is challenging . Importantly, supervised GEBD models are trained to detect fine-grained visual discontinuities—often frame-level motion changes, cut-points, or small appearance shifts. This objective is intentionally much finer than the episodic segmentation problem we study, which aims to identify coarse, meaningful transitions rather than dense micro-boundaries. As noted by Rai et al. (WACV 2023), “not every perceptual or motion change corresponds to a meaningful boundary,” and supervised GEBD systems tend to over-segment because they optimize for such micro cues.
> In contrast, unsupervised GEBD-style methods operate primarily through temporal sensitivity parameters (e.g., local-window similarity, temporal stride) and can be adapted to longer unstructured videos such as ADL-GEBD simply by adjusting these temporal hyperparameters. Our goal is not to benchmark against fully supervised GEBD models—which solve a different granularity of the problem—but to provide a label-efficient boundary detector that remains stable, avoids over-segmentation, and offers refined, semantically stronger candidate episodes.
> Our abstract intended this only as motivation: prior embodied-AI work has shown that navigation and search improve when memory emphasizes stable, scene-level transitions rather than frame-level jitter (e.g., the Scene Memory Transformer, Fang et al., CVPR 2019, which uses farthest-point sampling to retain meaningful state changes). Our method provides an explicit way to obtain such coherent units. In addition, Section 5.2 describes moment queries as a downstream task that naturally benefits from boundary-level representations of this kind. We have also included, in the appendix, a QA evaluation on memory retrieval using the events stored by our method. This further illustrates how these structured event boundaries support downstream reasoning tasks, even though embodied-agent experiments are not part of this paper.
> [1]Rai, A. K., Krishna, T., Dietlmeier, J., McGuinness, K., Smeaton, A. F., & O’Connor, N. E. (2022). Motion Aware Self-Supervision for Generic Event Boundary Detection. arXiv:2210.05574.
> [2]Rabe, F., & Wachsmuth, I. (2012). Cognitively Motivated Episodic Memory for a Virtual Guide. In ICAART (pp. 524–527).
> [3]Fang, K., Toshev, A., Fei-Fei, L., & Savarese, S. (2019). Scene Memory Transformer for Embodied Agents in Long-Horizon Tasks. arXiv:1903.03878.
>
>
>
> 3) Regarding weekness point 3, we acknowledge that CLIP, DINOv2, captioning, and scene graphs are standard components in video understanding. Our contribution lies in the careful integration of these modalities, guided by weighted analysis to determine the optimal combination and aggregation level for producing episodic-level boundaries. This analysis ensures that each modality contributes effectively to reducing micro-level transitions and consolidating semantically coherent segments, which prior multimodal GEBD approaches do not address. Ablation and weighting studies (Sec. 5.3.4) confirm that deviating from this integration leads to less stable boundary predictions, highlighting the practical advantage of our design.

---

> ### Author Response · Authors · 2025-11-24
>
> Regarding Weakness Point 4, we have improved the flow and clarity of several sections in the paper. In addition, Figure 1 has been updated to correct the previously misplaced label: “uncertainty-driven consolidation” (which was misspelled and mistakenly placed under Level 1) is now correctly positioned under Level 2.
>
> We believe the reviewer’s concern may stem from a misunderstanding of the distinction presented in Table 1. The distinction we make is accurate: although both GEBD and our task involve detecting event boundaries, the types of boundaries they target differ fundamentally. As explained in Table 1—and as stated by the GEBD authors themselves—the Kinetics-GEBD benchmark is designed for fine-grained, micro-level boundaries within short clips, capturing subtle transitions such as “holding the cup handle” vs. “lifting the cup.” These serve a different application space.
>
> In contrast, our task focuses on scene- and situational-level boundaries, such as transitions between rooms (e.g., kitchen → living room), which naturally arise in longer, ADL/Ego-style videos. The GEBD authors explicitly note (Section 9.2, Supplementary) that long videos predominantly involve scene-level changes, which differ substantially from the micro-transitions emphasized in Kinetics-GEBD.
>
> [1] Generic event boundary detection: A benchmark for event segmentation. ICCV 2021. [2] Intra-and inter-action understanding via temporal action parsing. CVPR 2020.

---

> > ### Comment · Reviewer_2kfw · 2025-11-27
> >
> > I sincerely thank the authors for their comment, which resolved the confusion regarding GEBD vs. the paper's task. However, I still have major concerns:
> >
> > - The paper's clarity is still too low (e.g., missing method details, clumsy reference formats)
> > - Contributions are not validated enough. For example, it is misleading to claim to beat heavily supervised models in the abstract, then only compare against unsupervised methods in Table 2.
> > - The method (combining multimodal encoders with learned adaptive thresholding, initialized with empirically found values) is not novel or original enough to be accepted in this venue.
> >
> > I have updated my score from 0 to 2 as some of my concerns resolved, but I still believe the paper requires major improvements and another round of review.

---

> > > ### Author Response · Authors · 2025-11-28
> > >
> > > The claim in the abstract refers to our Ego4D results (Table 3), where we directly outperform heavily supervised moment-localization models.
> > > Table 2 is different: it uses ADL-GEBD, where our adaptive threshold was not trained. The threshold is trained only on Ego4D, then applied zero-shot to ADL. Because we do not use ADL supervision, we compare only against unsupervised/self-supervised GEBD baselines there.
> > > We have clarified this in the paper to avoid confusion.We would also like to clarify that although Level 1 includes a light, sparsely-supervised threshold-calibration module, the boundary detector itself operates in a semi-supervised manner and does not use dense frame-level annotations. Its behavior is much closer to unsupervised novelty detectors than to supervised GEBD classifiers. Therefore, as shown in Appendix C, comparing Level-1 outputs with unsupervised baselines is methodologically appropriate because (i) the learned module only calibrates sensitivity rather than learning boundary semantics, and (ii) the core detection logic (backward similarity + keyframe-based novelty) is label-free.
> > >
> > > Our novelty does not lie in merely combining encoders with thresholding. The key contribution is a backward-only, consolidation-driven boundary detector that revisits past keyframes rather than relying on instantaneous frame differences, along with a light, scenario-adaptive threshold module that calibrates sensitivity without learning boundary semantics. Coupled with our second-level episodic consolidation, these components form a hierarchical segmentation framework that has not previously been explored. This design introduces a fundamentally different way of structuring continuous experience into episodic units

---

### Author Response · Authors · 2025-11-24

Dear Reviewers,

We thank you sincerely for your constructive and thoughtful feedback. We have carefully revised the paper and uploaded an updated PDF reflecting the changes based on your comments. In this revision, we have added  additional results, included a few downstream evaluations in the appendix, and clarified multiple sections to better address the concerns raised.

We appreciate your time and insights, which have significantly improved the quality and clarity of the work.

---

### Meta-Review · Area_Chair_qpG6 · 2026-01-09

**Summary:**

This paper introduces a two-level, cognitively inspired event segmentation scheme for unstructured video. The first level uses a semi-supervised adaptive thresholding detector that filters out noise and repetitive micro-actions. The second level retrospectively validates and merges boundaries using semantic, perceptual, and audio cues, yielding stable episodes grounded in meaning rather than transient visual changes. The reviewers appreciated that the proposed two-level architecture (detection, consolidation) is intuitive and well-motivated. Segmenting semantically coherent episodes is indeed meaningful and aligns with real-world applications. However, they raised concerns regarding limited novelty, insufficient validation, unclear/ambiguous explanation of the proposed method, and writing/presentation issues including wordings. The authors tried to address the raised concerns with providing additional experiments in their rebuttal and revised paper.  AC thinks that some were resolved while some were not fully resolved.

Regarding the novelty, the authors argued the difference between GEBD and the task on which this paper focuses, and carefully explained the contribution of this paper.  They also corrected misunderstandable word-usage. These clarified real novelty of this paper to some extent. However, combining multimodal encoders with learned adaptive thresholding, initialized with empirically found values is not ground-breaking innovation. Although the introduced task itself seems interesting, the technical novelty to solve the task is incremental.

Regarding the validation of the proposed method, the authors provided additional experiments to address the raised concerns.  The authors made efforts to convince the comparison with unsupervised methods; however, seemed to fail.  Even though the boundary detector operates in a semi-supervised manner and does not use dense frame-level annotation, and, furthermore, the behavior is much closer to unsupervised novelty detectors, it is not fair by nature to compare supervised models with unsupervised models.  In order to justify such comparison, more rigorous justification is required.  In addition, sensitivity against the choice of the pre-trained feature extractor is unaddressed.  Evaluation on different types of datasets such as sorts or surveillance is also unaddressed. The weaknesses pointed by ReviewerqYeb are unaddressed.

The explanation of the proposed method was improved; however, writing issues still remain. For instance, even though a new experimental result is added in Appendix B to demonstrate broad applicability, no argument is provided in the paper to position this result, meaning not solidly rebutted.  The rebuttal itself is also too vaguely descriptive but not concise; it should have been more specific to the point.  AC thinks that writing should be further improved to properly support claims in the convincing way.

AC thinks that the task this paper focuses on is reasonable and worth exploring, and the proposed method is interesting; however, more thorough analysis/validation of the proposed method to support the claims in the paper should be required.  In addition, writing and presentation should be improved. AC thinks that the remaining concerns outweigh the technical contribution of this paper; the paper will be beneficial from substantial improvement.  For this, this paper cannot be accepted.  Note that clarifying revised parts with highlights in the revised manuscript is kind to reviewers and AC.

**Reviewer Concerns:**

Main concerns are on limited novelty, insufficient validation, unclear/ambiguous explanation of the proposed method, and writing/presentation issues including wordings.  As written above, these concerns were not fully addressed, leaving remaining concerns.

**Reviewer Scores:**

Reviewer 2kfw would raise his/her score from 0 to 2 as some of his/her concerns were resolved while he/she thinks that major improvement is still required. Reviewer qYeb would keep his/her initial score or degrade the score to 6 because his/her concerns written in Weaknesses were unaddressed; only the questions were addressed. Reviewer dy9s would keep his/her initial score because most of his concerns seems resolved but not fully. Reviewer TqQb would keep his/her initial score.  This is because concern on insufficient validation is mostly unaddressed and the task itself is not attractive broadly. Reviewer CmrE would raise his/her score from 6 to 8 because additional explanations and experiments provided in the rebuttal have convinced him/her.

---

### Decision · Program_Chairs · 2026-01-26

Reject